# Variables Influencing the Pressure and Volume of the Pulmonary Circulation as Risk Factors for Developing High Altitude Pulmonary Edema (HAPE)

**DOI:** 10.3390/ijerph192113887

**Published:** 2022-10-26

**Authors:** Nina Hundt, Christian Apel, Daniela Bertsch, Carina Cerfontaine, Michael van der Giet, Simone van der Giet, Maren Graß, Miriam Haunolder, Nikole M. Heussen, Julia Jäger, Christian Kühn, Audry Morrison, Sonja Museo, Lisa Timmermann, Knut Wernitz, Ulf Gieseler, Thomas Küpper

**Affiliations:** 1Institute of Occupational & Social Medicine, RWTH Aachen Technical University, 52074 Aachen, Germany; 2Department of Operative Dentistry, Periodontology & Preventive Dentistry, RWTH Aachen Technical University, 52074 Aachen, Germany; 3Institute of Applied Medical Engineering, RWTH Aachen University, 52074 Aachen, Germany; 4Department of Cardiology, Catholic Hospital Marienhof, 52074 Koblenz, Germany; 5Department of Medical Statistics, RWTH Aachen Technical University, 52074 Aachen, Germany; 6Center of Biostatistics and Epidemiology, Medical School, Sigmund Freud Private University, 1020 Vienna, Austria; 7Medical Commission of the Union Internationale des Associations d’Alpinisme (UIAA MedCom), CH-3000 Bern, Switzerland; 8Royal Free London NHS Foundation Trust, London NW3 2QG, UK

**Keywords:** HAPE, risk factors, echocardiography, pulmonary pressure, workload

## Abstract

Background: At altitudes above 2500 m, the risk of developing high altitude pulmonary edema (HAPE) grows with the increases in pulmonary arterial pressure. HAPE is characterized by severe pulmonary hypertension, though the incidence and relevance of individual risk factors are not yet predictable. However, the systolic pulmonary pressure (SPAP) and peak in tricuspid regurgitation velocity (TVR) are crucial factors when diagnosing pulmonary hypertension by echocardiography. Methods: The SPAP and TVR of 27 trekkers aged 20–65 years en route to the Solu Khumbu region of Nepal were assessed. Echocardiograph measurements were performed at Lukla (2860 m), Gorak Shep (5170 m), and the summit of Kala Patthar (5675 m). The altitude profile and the participants’ characteristics were also compiled for correlation with the measured data. Results: The results showed a highly significant increase in SPAP and TVR after ascending Kala Patthar. The study revealed a lower increase of SPAP and TVR in the group of older participants, although the respective initial measurements at Gorak Shep were significantly higher for this group. A similar finding occurred in those using Diamox^®^ as prophylaxis. There was an inverse relationship between TVR and SPAP, the peripheral capillary oxygen saturation, and heart rate. Conclusions: The echocardiograph results indicated that older people are an at-risk group for developing HAPE. A conservative interpretation of the basic tactical rules for altitudes should be followed for older trekkers or trekkers with known problems of altitude acclimatization (“slow acclimatizer”) as SPAP elevates with age. The prophylactic use of Acetazolamide (Diamox^®^) should be avoided where not necessary for acute medical reasons. Acetazolamide leads to an increase of SPAP, and this may potentially enhance the risk of developing HAPE. Arterial oxygen saturation measurements can provide an indicator for the self-assessment for the risk of developing HAPE and a rule of thumb for the altitude profile, but does not replace a HAPE diagnosis. Backpack weight, sex, workload (actual ascent speed), and pre-existing diseases were not statistically significant factors related to SPAP and TVR (*p* ≤ 0.05).

## 1. Introduction

Since the first successful ascent of Mount Everest in 1953, the base camp at its foot has become a point of global attraction. Each year more people come to the Solu Khumbu region in Nepal to undertake the eight-day trek to the base camp. In 1979, only about 3600 tourists made it there. This number steadily increased from 32,124 in 2010 to more than 57,000 in 2019 [1]. The combined number of trekking and mountaineering visitors to Nepal in 2006 was 383,926 and rose to 1,197,191 in 2019 [1]. Over the last 20 years, an increasing number of these trekkers are older people [1] or have pre-existing illnesses presenting a specific risk profile that increases the probability of suffering serious health complications due to altitude or even death [2,3,4,5,6]. While the pathophysiology of high-altitude pulmonary oedema (HAPE) is relatively clear, little is known about the risk factors. Scant data makes statements about the individual risk profile and effective prevention difficult. In addition, few studies have been conducted under real hypoxic conditions at high altitudes (hypobaric hypoxia), and most of them suffer from small sample sizes.

In a retrospective study, Leshem et al. analysed the incidence of illnesses related to extreme altitude in 406 patients who were treated at the CIWEC clinic in Kathmandu, Nepal, from 1999 to 2006 [7]. They found that 21% had suffered high-altitude cerebral oedema (HACE), 34% a HAPE, and 27% a combination of both. They also found that 18% were evacuated and treated in Kathmandu for acute mountain sickness (AMS). Although AMS is generally not indicated for a helicopter evacuation, the number of trekkers being flown out of the Everest region has considerably increased. In 1999, 56 out of 100,000 trekkers had to be evacuated, but by 2005 the respective number rose to 276. This may reflect the higher incidence of emergencies coming with increased tourism, but there were also improvements to technical possibilities for evacuation over this time. The annual cost of flying trekkers into lower altitudes is about USD225,000.

The number of fatal incidents among 100,000 trekkers in the Everest region increased from 2.02 (1984–1987) to 3.62 (1987–1991) and has subsequently risen significantly to 7.7 [7]. The average age of trekker fatalities went up as well from 35 ± 13 years from 1984–1987 to 50.9 ± 13.7 years from 1999–2006.

The actual study was enrolled during the ADEMED Expedition (Aachen Dental and Medical Expedition) to Mt. Everest and sought to verify changes in pressure and volume of the pulmonary circulation during acute exposure to high altitude and physical stress, together with possible factors influencing HAPE.

Specifically, our aims were threefold:

To document the development of SPAP and TVR by echocardiography under real conditions during ascent to extreme altitude (up to 5,600 m), with and without additional weight load.

To identify or exclude possible factors influencing SPAP and TVR.

To develop more specific recommendations for preventing people from suffering HAPE during ascent to extreme altitudes.

## 2. Materials and Methods

The research was designed as a prospective cohort study. For the echocardiographic analysis, three measuring station locations were selected at different altitudes: Lukla (2860 m), Gorak Shep (5170 m), and Kala Patthar (5675 m). Lukla is the gateway to the Solu Khumbu region and the point of departure for all trekkers aiming for the Everest Base Camp. As the next road is seven days away by foot, most trekkers fly into Lukla from Kathmandu (1355 m) on small twin-engine propeller aircraft. Most trekkers reach the second measuring point at Gorak Shep (5170 m) after seven days of hiking. From there, the Everest Base Camp is just a one-day hike away. The third measuring point was at Kala Patthar (5675 m), which many trekkers climb to view the breath-taking sunrise across the Everest range before going on to the Everest Base Camp. The hike from Gorak Shep to Kala Patthar is not challenging and takes about two hours on average to ascend another 500 m of altitude.

The study sample consisted of trekkers following the Everest Base Camp Trek. The majority of participants were recruited from lodges in Gorak Shep. After being orally informed about the objectives of the study, the volunteers gave written consent. Two sessions were arranged to collect measurements and data for each participant.

Inclusion criteria required the trekker to be aged at least 18 years, without language barriers, and with unrestricted judgement capacity. Criteria for exclusion were relevant deviations from the regular altitude profile and a chronic obstructive pulmonary disease (COPD) of the Global Initiative for Chronic Obstructive Lung Disease (GOLD) stage 4.

At each measuring point, the examination took place under thermal well-being. At Kalar Patthar, a small 3-person tent pitched close to the summit provided shelter. The echocardiograph examination was always carried out by the same experienced investigator.

The first measurement was taken in the evening after arriving at Gorak Shep (5170 m). The second measurement was taken the next morning after reaching the summit of Kala Patthar (5675 m), directly after a pressure phase with increasing hypobaric hypoxia. Each measurement consisted of an echocardiographic examination of the right heart and readings of blood pressure, heart rate, and oxygen saturation. In addition, the lungs were auscultated, and the participant was checked for peripheral oedema, jammed jugular vein, and possible signs of ataxia. The weight of the daypack was measured at Kala Patthar using a calibrated spring scale. This weight was correlated with each participant’s body weight. The trek duration to reach the summit of Kala Patthar was also recorded. Every participant filled in a questionnaire including questions on the altitude profile, pre-existing diseases, and their current medication.

The measurement procedure was conducted with two cohorts: (a) Collective Lukla–Gorak Shep–Kala Patthar and (b) Collective Gorak Shep–Kala Patthar, which included Collective (a). As the number of participants who were examined in Lukla was relatively small (7 participants) and only included the members of the ADEMED-Expedition, this study focused only on the collective Gorak Shep–Kala Patthar. Three of these participants ascended Kala Patthar several times. In this case, only the first ascent was considered to avoid deviations due to acclimatisation. The remaining data was summarized in time series.

For the echocardiographic examination, a portable ultrasound device type Vivid-I™ from GE Healthcare (General Electrics, Boston, MA, USA) was used. This device is the size of a laptop and is equipped with a sector probe and internal data storage. The power supply was provided through the internal accumulator and a gasoline-powered generator (G1000i by Changzhou ATIMA Power Machines Co., Ltd., Ankang, Shaanxi, China). The echocardiographic images were interpreted on-site. The SPAP, TVR, and the size and collapsibility of the inferior vena cava are needed to assess pulmonary pressure [8]. The echocardiographic data was interpreted according to the European guidelines for diagnosis and therapy of pulmonary hypertension (Table 1).

For analysing the oxygen saturation, a portable pulse oximeter was used (Contec Medical Systems CMS-50E, Qinhuangdao, China). Blood pressure was determined by using a Visomat Comfort E cuff (Uebe Medical GmbH, Wertheim, Germany).

The study was in full accordance with the Declaration of Helsinki [9] (WMA 1964) and counselled by the Ethical Commission at RWTH Technical University, Aachen, Germany (Study No. EK196/11).

A descriptive statistical analysis was applied with non-parametric tests (Chi-square test and Mann-Whitney U-test). For verifying a linear relation, Pearson’s correlation coefficient was determined [10] with *p* < 0.05 defined as significant. An adequate correlation is the basic requirement for a causal relation of two variables but does not allow a final statement concerning a cause-and-effect relationship. In these cases, regression analysis with one dependent and one independent variable was used [11]. The evaluation was conducted on the basis of different factors: the coefficient of determination (R^2^), standard error of the estimator, an f-test, and t-test [10] with *p* < 0.05 was defined as significant, and *p* < 0.1 as a tendency. If more than one factor influencing SPAP and TVR was considered, a multiple regression analysis was needed. In this case, the two factors with the highest bivariate correlation were included in the analysis—the participant’s age and heart rate [11].

## 3. Results

The 27 male and female participants (aged 20–65 years) were generally a normal weight and a healthy collective (mean BMI: 22; range 19–27). Only three participants suffered from arterial hypertension, one had mild pre-existing pulmonary hypertension, and another had mild chronic obstructive pulmonary disease, which was well stabilized by drug treatment. The altitude profile of all participants was almost identical. Only a minority had been exposed to altitudes over 3000 m previously.

In nearly all participants, an increase in SPAP and TVR could be detected at altitude. On average, during the ascent of Kala Patthar, the SPAP rose by 4.39 mmHg and TVR by 0.22 m/s. These differences were significant (*p* < 0.01). Evaluating the data based on the criteria of the European guidelines for diagnosing pulmonary hypertension [8], in 17 participants (63%), pulmonary hypertension was improbable in Gorak Shep (Figure 1). On Kala Patthar, pulmonary hypertension was improbable for only six participants (22%) (*p* < 0.01). For 21 participants (78%), pulmonary hypertension in Kala Patthar was probable, compared with only ten (37%) participants in Gorak Shep (*p* < 0.01). This meant that the risk of developing pulmonary hypertension doubled during a straight ascent from 5170 m to 5675 m.

The participant’s age was a significant factor influencing SPAP and TVR in Gorak Shep in the regression analysis (Figure 2 and Figure 3). This also applied to the comparison of mean values of two subgroups divided according to their age: <50 years (*n* = 14) and >50 years (*n* = 13). The SPAP average in Gorak Shep was 35.9 mmHg for those <50 years and 42.7 mmHg for those >50 years. This difference was significant (*p* < 0.05) (Figure 2). On Kala Patthar, no significant difference was found between these two age groups related to TVR or SPAP.

The results of the measurements in Gorak Shep showed a low correlation between the peripheral capillary oxygen saturation, SPAP, and TVR (TVR: r = −0.227; SPAP: r = −0.227). Similar results were found on Kala Patthar (TVR: r = −0.211; SPAP = −0.14). In addition, a positive linear correlation between the total increase of SPAP, TVR, and peripheral capillary oxygen saturation was detected (TVR: r = 0.356; SPAP: r = 0.269). However, the regression analysis did not show this relationship to be significant.

With regards to the number of days needed for ascending from Lukla and the level of SPAP or TVR, no relationship could be detected. Nevertheless, the time series of the three participants who ascended Kala Patthar several times (up to 12×) showed that after a few days, the measured data did not differ from the data recorded in Lukla.

Even though no significant differences were found in TVR and SPAP for participants with pre-existing lung diseases in Gorak Shep or on top of Kala Patthar, the absolute increase of SPAP differed significantly between those with and without pre-existing lung diseases (*p* < 0.05). The average absolute increase in those without pre-existing lung diseases was 4.96 mmHg (±7.2) compared to −6.15 mmHg (±3.08) in those with pre-existing lung diseases.

Eight of twenty-seven participants had taken acetazolamide (Diamox^®^ or similar products). Four had used it as prophylaxis, three as a demand medication, and one for both these reasons. The dose specification ranged between 125 to 250 mg twice a day as prophylaxis and 250 mg for demand medication. The measured results in Gorak Shep proved that acetazolamide taken prophylactically had a significant impact on the height of SPAP (*p* = 0.05) as it was higher. Without acetazolamide, the average value in Gorak Shep was 37.0 mmHg (±6.09) and 43.8 mmHg (±5.13) on top of Kala Patthar. In contrast, the average value with acetazolamide as prophylaxis was 45.4 mmHg (±13.25) in Gorak Shep (*p* < 0.05) and 40.7 mmHg (±11.71) on Kala Patthar *p*(SPAP) > 0.05.

Significant differences could also be observed in the increase of TVR and SPAP, comparing the measurements in Gorak Shep and Kala Patthar (*p*(TVR) < 0.05 and *p*(SPAP) < 0.05, respectively). The measured values of TVR, as well as SPAP from participants with acetazolamide prophylaxis, were lower than those from trekkers without such prophylaxis. In the group of trekkers without acetazolamide prophylaxis, the average increase of TVR was 0.3 m/s (±0.29) and 6.8 mmHg (±6.12) in the case of SPAP, respectively. In contrast, the measured values of participants with acetazolamide prophylaxis revealed an average decrease of TVR of −0.07 m/s (±0.22) and PAP of −4.7 mmHg (±4.07).

The collected data did not glean any further insight which might have influenced the height or level of increase of TVR and SPAP from the other variables assessed: sex, body mass index, speed of the ascent, or the weight of the backpack.

The result of multiple regression analysis showed significant linear regression in Gorak Shep: SPAP = 62.665 + 0.308x_1_ − 0.422x_2_ where x_1_ was age, and x_2_ was heart rate. The f-test also confirmed both regression coefficients were significant (*p* < 0.05). The coefficient of determination was 0.327. The inclusion of more factors did not show any significant results and only increased the coefficient of determination.

The results can be summarized as follows: With increasing altitude, TVR and SPAP increase. Staying at the same altitude, as well as applying the principle of “climb high, sleep low”, seemed to influence TVR and SPAP positively, and after a while, the values recorded in Lukla were reached again.

Data indicated that older people had a higher risk of developing HAPE. With increasing age, SPAP, as well as TVR, rose, even if a significant relationship could not be detected. There was a low inverse correlation between the SpO_2_ and TVR. The higher the SpO_2_, the lower the TVR. However, a statistically significant relationship was not detected. The collected data did not provide any evidence that sex, BMI, or the weight of the backpack influenced the measured values or increased SPAP and TVR.

Even if the number of affected persons using acetazolamide as prophylaxis was small, their initial SPAP values in Gorak Shep were significantly higher (Figure 4). In addition, acetazolamide influenced the absolute increase of SPAP and TVR while ascending Kala Patthar (Figure 5). In this group, the increase was smaller compared with those participants who did not take Diamox ^®^ as prophylaxis.

## 4. Discussion

In 2011 more than 736,000 tourists travelled to Nepal, of which about 35,000 visited the Sagarmatha National Park. These tourists (52% women; 48% men) were aged between 30 and 45 years. In 2019, this visitor number increased to more than 57,000 with a similar sex and age distribution [1]. Our study’s demographic for age and sex was also quite similar to this. So far there has been only one year (2011) when women outnumbered men (52.2%, [1]).

Sixty percent of HAPE cases mainly occur between 3500 and 6000 m (survey in [12]). The risk of developing HAPE increases with altitude (especially when this ascent was critically fast), together with pulmonary hypertension. Hypoxic vasoconstriction (“Euler-Liljestrand reflex”) leads to an increase in SPAP, which can culminate in pulmonary hypertension. This statement is supported by several other studies. Maggiorini et al. found that an increase in SPAP was responsible for the developing HAPE and not the higher permeability of the pulmonary vessels [13]. Pagé et al. found that with rising altitude, the pressure in the pulmonary circulation increased, and subclinical pulmonary oedema was the normal case [14]. Therefore, HAPE is the result of a progressive increase in pressure and does not follow an all-or-nothing principle. In the present study, an increase in TVR and SPAP was also found. During the ascent from Gorak Shep (5170 m) to the top of Kala Patthar (5675 m), a significant increase in SPAP and TVR was documented. These results were supported by the increase in the likelihood of developing a HAPE on Kala Patthar that was detected (Figure 1). Accordingly, the percentage of participants with statistically improbable HAPEs declined from 63% in Gorak Shep to 22% in Kala Patthar.

Our study showed a statistically significant regression between age and SPAP. According to the statistical evaluation, SPAP rises by 0.748 mmHg per year of life. The multivariable regression analysis resulted in an increase of 0.308 mmHg per life year. The comparison of mean values of the collective >50 years and <50 years resulted in significant results as well. These results corresponded to the findings of Basnyat et al., who described an increase of <1 mmHg per life year [15]. The descriptive study of Leshem et al. concerning the incidence of patients suffering from HAPE between 1999 and 2006 also supported these findings. The group of patients suffering from HAPE was older than those who were suffering from AMS or HACE (HAPE: 48 ± 12 years; HACE: 44 ± 14.1 years; HACE + HAPE: 43 ± 12.8 years; AMS: 41 ± 14.5 years; control group: 38.6 ± 13.9 years) [7]. A possible reason could be the increasing atherosclerotic changes with age [16]. At the same time, the elasticity of the vessels and, therefore, the adaptive capacity to respond to changes in blood pressure decreased. Another explanation could be a reduced fluid intake of the elderly people that leads to an increase of blood viscosity and to a sympathetic counter-regulation and consequently to increasing pressure at high altitudes. The compensation for these additional stress factors may influence the level and the increase in SPAP and TVR. The higher standard deviation among the subgroup of older participants supports this theory. The missing age group between 35 and 49 years could be a possible bias.

In the retrospective study of Leshem et al., 136 cases of HAPE during the years 1999 to 2008 were documented [7]. The proportion of males suffering from HAPE was significantly higher (56.6%) than the control group of females. According to this data, sex would have to be considered as a possible risk factor for developing a HAPE. However, our study did not support this finding. It is possible that Leshem’s data included a bias because, to the best of our best knowledge, his statistics were not corrected for the gender of the total collective of trekkers, where males significantly predominate except in 2011 [1].

Basnyat et al. discovered an inverse linear relationship between arterial oxygen saturation and SPAP [15]. The increase of 1% of the arterial oxygen saturation led to a decrease of about −0.31 mmHg (*p* < 0.01). Even if it was not significant, an inverse relation was also described in the study of Baggish et al. [17]. In the actual study, a weak inverse correlation between TVR and arterial oxygen saturation could be detected, but these results were not significant. Even if the results were not significant, a tendency could be described. With decreasing arterial oxygen saturation, the pressure in the pulmonary circulation increases. Even though pulse oximeters tend to have a higher error rate, they can be used under standardized conditions (warm extremities, always the same finger in the same position, etc.) for evaluating the course of arterial oxygen saturation over several days. If the course deviations from the rest of the group members differ considerably, this can be used as an early warning system for HAPE. Nevertheless, pulse oximeters can never replace a clinical diagnosis.

Acetazolamide (Diamox^®^) is very popular for preventing AMS. According to the current data, the prophylactic intake of the drug affects the pressure of the pulmonary circulation, especially the SPAP. It leads, on the one hand, to a higher initial value of SPAP in Gorak Shep but, on the other hand, to a lower absolute increase of SPAP and TVR during ascent. A possible explanation may be the diuretic effect of acetazolamide which aggravated the dehydration by altitude diuresis [12]. The body is not able to compensate for the loss of fluid anymore (e.g., by a fluid shift from intracellular compartments), and with the resultant increased blood viscosity, the activity of the sympathetic nervous system rises. The heart frequency increases, the stroke volume decreases, and the vasoconstriction of the vessels was followed by an increase of the pressure in the pulmonary circulation. However, since the preventive effect of acetazolamide is obviously independent of a wide range of its dosage [18,19], but the diuretic side effect is dose-dependent, perhaps this specific hypothesis can be investigated in the future.

Similar to the collective of older participants, the participants who took acetazolamide as prophylaxis showed a significantly lower absolute increase of SPAP and TVR than the control group. A reason for that could be that the already high activity of the sympathetic nervous system cannot be enhanced or can only do so slowly. This statement is limited by the small number of participants who took acetazolamide prophylactically and the study design that did not include direct data acquisition concerning the sympathetic system.

The effect of up to 12 ascents to the summit of Kalar Patthar on the factors measured is an interesting observation. Due to the small number of participants, this must be interpreted with care, but it may be a consequence of an increase in acclimatization time. For a period of about two weeks, an increase in ventilation (“ventilatory acclimatization”) causes an increase of SaO_2_ of about 6% [20]. This should significantly open the pulmonary vessels by the Euler-Liljestrand reflex and reduce pulmonary pressure. Again this may be investigated by a specific study of subjects who ascend to altitude and stay there for several days, preferably without any significant exercise.

## 5. Conclusions

The echocardiographic results indicate that older people belong to a risk group for developing HAPE. The prophylactic use of acetazolamide led to higher initial SPAP values in Gorak Shep and a lower increase of PAP as well as TVR after ascending Kala Patthar. Based on these results, the following recommendations can be made:Follow the basic tactical rules for altitude ascent to reduce the risk and even prevent a HAPE. A well-conceived altitude profile (increase sleeping altitude for not more than about 300 m per day above 2500 m) reduces SPAP and TVR.Conservative interpretation of the basic tactical rules for altitudes should be followed for older trekkers or trekkers with known problems of altitude acclimatization (“slow acclimatizer”) as SPAP is elevated with age. That does not impede staying at high altitudes, but these people should be advised to be aware of the risk of overburdening. A slow ascent with extra days for recovery and acclimatization can help the body to adapt to the altitude and the SPAP to decrease to the initial values. The risk of developing a HAPE can consequently be minimized.Avoid taking Diamox^®^ when not necessary for acute medical reasons. Acetazolamide leads to an increase in SPAP, and this may potentially enhance the risk of developing HAPE.The arterial oxygen saturation may serve as an indicator for self-assessing the risk of developing a HAPE and provides a thumb rule for the altitude profile.

## Figures and Tables

**Figure 1 ijerph-19-13887-f001:**
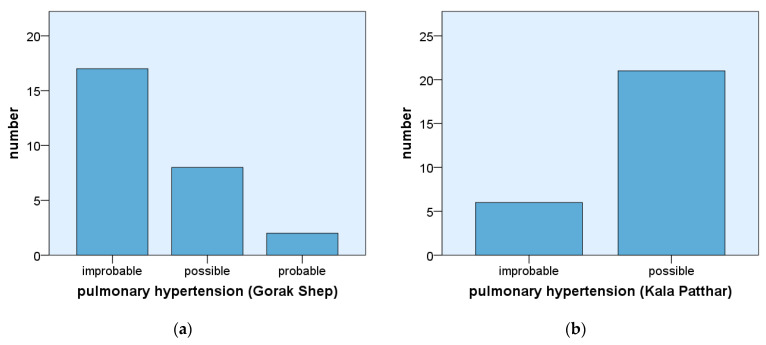
Pulmonary hypertension of participants at Gorak Shep (**a**) and at the summit of Kala Patthar (**b**) (*n* = 27).

**Figure 2 ijerph-19-13887-f002:**
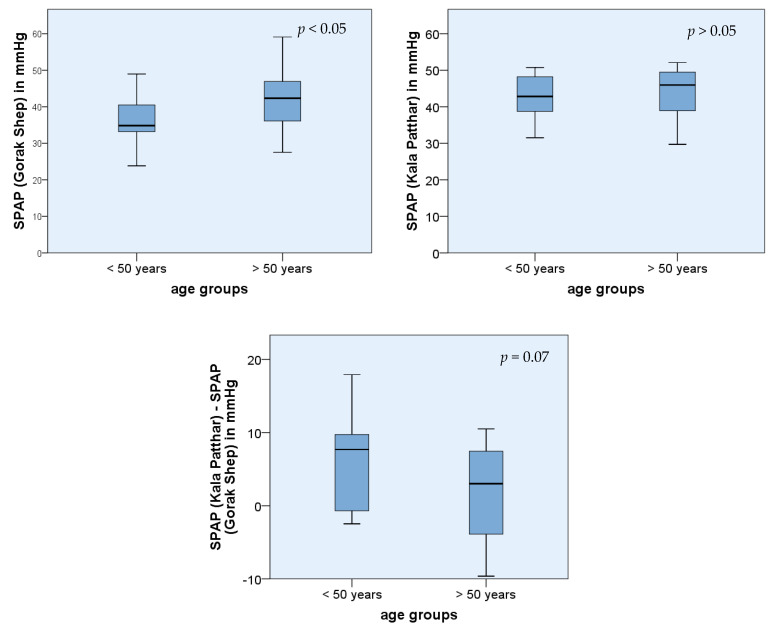
Comparison of mean values of age subgroups (<50 years (*n* = 14), >50 years (*n* = 13)).

**Figure 3 ijerph-19-13887-f003:**
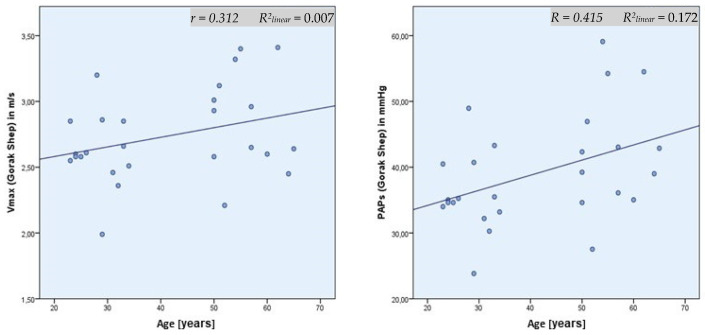
Scatter plot of age and Vmax (**left**; r = 0.312, R^2^ = 0.007) and PAPs (**right**; r = 0.415, R^2^ = 0.172); data from Gorak Shep (5140 m).

**Figure 4 ijerph-19-13887-f004:**
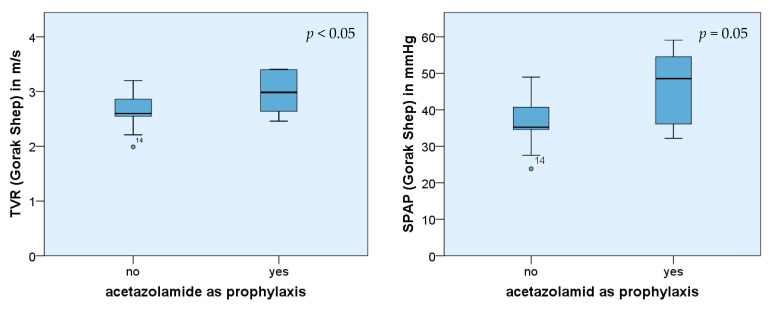
Comparison of mean values and SPAP of TVR in Gorak Shep: subgroup 1 with acetazolamide as prophylaxis (*n* = 4), subgroup 2 without intake of acetazolamide (*n* = 21).

**Figure 5 ijerph-19-13887-f005:**
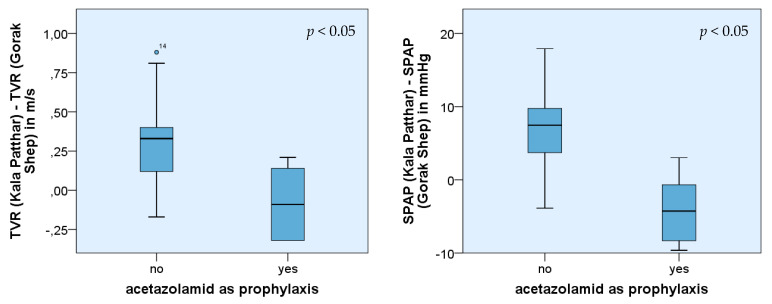
Comparison of mean values of TVR and SPAP after ascent from Gorak Shep to Kala Patthar of two subgroups: subgroup 1 with acetazolamide as prophylaxis (*n* = 4), subgroup 2 without intake of acetazolamide (*n* = 21).

**Table 1 ijerph-19-13887-t001:** Evaluation criteria for pulmonary hypertension by using echocardiography [8].

**Diagnosis by Echocardiography: Pulmonary Hypertension Improbable**
TVR ≤ 2.8 m/s, SPAP < 36 mmHg with no other signs of pulmonary hypertension
**Diagnosis by echocardiography: Pulmonary hypertension possible**
TVR ≤ 2.8 m/s, SPAP < 36 mmHg, but there are other signs of pulmonary hypertension TVR 2.9–3.4 m/s, SPAP 37–50 mmHg with no other signs of pulmonary hypertension
**Diagnosis by echocardiography: Pulmonary hypertension probable**
TVR > 3.4 m/s, SPAP > 50 mmHg, and no other signs of pulmonary hypertension

## Data Availability

Not applicable.

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
