# Peer review of "Variables Influencing the Pressure and Volume of the Pulmonary Circulation as Risk Factors for Developing High Altitude Pulmonary Edema (HAPE)"

_ijerph, 2022, doi:10.3390/ijerph192113887_

Round 1

Reviewer 1 Report

The authors presented an interesting article on the risk factors underlying high altitude pulmonary edema (HAPE).

Three major concerns with the current version of the article.

1. Authors need to somewhat improve word usage. I would suggest avoid using idioms since they do not translate well.

2. Article needs a better presentation of data: (1) scatter plots of the main relation (SPAP as a function of age and heart rate) and (2) consider plotting all data points not just bar graphs.

3. Conclusions at the end of article do not match conclusions in the abstracts. I consider the conclusions at the end of the article more relevant and to the point.

Minor:

Table 1 uses European style for decimals (comma instead of the point) whereas the rest of the text uses North American convention

Author Response

Dear colleague,

we have included all your suggestions now in tha revised manuscript. Especially to add a scatter plot (now fig.3) was a very good idea. Thank you very much for your careful work!

ThK

Reviewer 2 Report

This is an interesting prospective cohort study on the potential variables the risk for developing high altitude pulmonary edema (HAPE). The study is interesting and well conducted. 

I suggest only some minor revisions

1)    Affiliations of the authors should be completed

2)    I suggest to remove “However” in the abstract background

3)    TRV and TVR are alternatively used, please use only one of these

4)    Explain better the first aim (line 76) in the Introduction

5)    Line 145 change were with was

6)    Line 155 remove were

7)    I suggest to remove commercial names of drugs

Author Response

Dear colleague,

we have included all your suggestions in the revised manuscript now. Thank you very much for your detailed and careful work!

ThK

Reviewer 3 Report

Congratulation, this is an interesting and well designed study! Within the conclusion section you state "A well-conceived altitude profile reduces SPAP and TRV". It would be fine, if you explain i litte bit more detailed, what you mean with this.

Author Response

(The authors gave the same response as above.)
